# Scaffold Repurposing Reveals New Nanomolar Phosphodiesterase Type 5 (PDE5) Inhibitors Based on Pyridopyrazinone Scaffold: Investigation of In Vitro and In Silico Properties

**DOI:** 10.3390/pharmaceutics14091954

**Published:** 2022-09-15

**Authors:** Kamelia M. Amin, Ossama M. El-Badry, Doaa E. Abdel Rahman, Magda H. Abdellattif, Mohammed A. S. Abourehab, Mahmoud H. El-Maghrabey, Fahmy G. Elsaid, Mohamed A. El Hamd, Ahmed Elkamhawy, Usama M. Ammar

**Affiliations:** 1Pharmaceutical Chemistry Department, Faculty of Pharmacy, Cairo University, Cairo 11562, Egypt; 2Pharmaceutical Chemistry Department, Faculty of Pharmacy, Ahram Canadian University (ACU), Giza 12566, Egypt; 3Department of Chemistry, College of Science, Taif University, Al-Haweiah, Taif 21944, Saudi Arabia; 4Department of Pharmaceutics, College of Pharmacy, Umm Al-Qura University, Makkah 21955, Saudi Arabia; 5Department of Pharmaceutics and Industrial Pharmacy, College of Pharmacy, Minia University, Minia 61519, Egypt; 6Department of Pharmaceutical Analytical Chemistry, Faculty of Pharmacy, Mansoura University, Mansoura 35516, Egypt; 7Biology Department, Science College, King Khalid University, Abha 61421, Saudi Arabia; 8Zoology Department, Faculty of Science, Mansoura University, Mansoura 35516, Egypt; 9Department of Pharmaceutical Sciences, College of Pharmacy, Shaqra University, Shaqra 11961, Saudi Arabia; 10Department of Pharmaceutical Analytical Chemistry, Faculty of Pharmacy, South Valley University, Qena 83523, Egypt; 11BK21 FOUR Team and Integrated Research Institute for Drug Development, College of Pharmacy, Dongguk University-Seoul, Goyang 10326, Korea; 12Department of Pharmaceutical Organic Chemistry, Faculty of Pharmacy, Mansoura University, Mansoura 35516, Egypt; 13Strathclyde Institute of Pharmacy and Biomedical Sciences, University of Strathclyde, 161 Cathedral Street, Glasgow G4 0NR, UK

**Keywords:** PDE5 inhibitors, pyridopyrazinone derivatives, scaffold repurposing, in vitro enzyme assay, 2D-QSAR analysis, molecular docking, molecular dynamic simulation

## Abstract

Inhibition of PDE5 results in elevation of cGMP leading to vascular relaxation and reduction in the systemic blood pressure. Therefore, PDE5 inhibitors are used as antihypertensive and antianginal agents in addition to their major use as male erectile dysfunction treatments. Previously, we developed a novel series of 34 pyridopyrazinone derivatives as anticancer agents (series **A**–**H**). Herein, a multi-step in silico approach was preliminary conducted to evaluate the predicted PDE5 inhibitory activity, followed by an in vitro biological evaluation over the enzymatic level and a detailed SAR study. The designed 2D-QSAR model which was carried out to predict the IC_50_ of the tested compounds revealed series **B**, **D**, **E** and **G** with nanomolar range of IC_50_ values (6.00–81.56 nM). A further docking simulation model was performed to investigate the binding modes within the active site of PDE5. Interestingly, most of the tested compounds showed almost the same binding modes of that of reported PDE5 inhibitors. To validate the in silico results, an in vitro enzymatic assay over PDE5 enzyme was performed for a number of the promising candidates with different substitutions. Both series **E** and **G** exhibited a potent inhibitory activity (IC_50_ = 18.13–41.41 nM). Compound **11b** (series **G**, oxadiazole-based derivatives with terminal 4-NO_2_ substituted phenyl ring and rigid linker) was the most potent analogue with IC_50_ value of 18.13 nM. Structure–activity relationship (SAR) data attained for various substitutions were rationalized. Furthermore, a molecular dynamic simulation gave insights into the inhibitory activity of the most active compound (**11b**). Accordingly, this report presents a successful scaffold repurposing approach that reveals compound **11b** as a highly potent nanomolar PDE5 inhibitor worthy of further investigation.

## 1. Introduction

Phosphodiesterase enzyme (PDE) plays an important role in regulating the intracellular level of cyclic guanosine monophosphate (cGMP), the second messenger of nitric oxide [1,2,3,4]. cGMP is hydrolyzed by phosphodiesterase enzyme into its inactive metabolite 5′-GMP which mediates a significant elevation of intracellular calcium concentration, resulting in vascular constriction [1,5]. Eleven subtypes of phosphodiesterase family have been described (PDE1–11) [6]. PDE5 is expressed mainly in vascular smooth muscle cells, lung, kidney, and platelets [5,7,8]. Inhibition of PDE5 results in elevation of cGMP and finally leads to vascular relaxation and reduction in systemic blood pressure [9,10]. Therefore, PDE5 inhibitors can be considered potential antihypertensive and antianginal agents. Moreover, the discovery of sildenafil (**1**, Viagra^®^, PDE5 inhibitor) revealed its potential use in the treatment of male erectile dysfunction [1,5,8,11,12]. Recently, it was approved for the treatment of pulmonary hypertension (Revatio^®^) [1,12]. In addition, it was reported for the treatment of cardiac hypertrophy [13,14]. Further structural optimization of sildenafil has driven the development of a new PDE5 inhibitor agent such as vardenafil (**2**, Levitra^®^) [1,7,15] and tadalafil (**3**, Cialis^®^) [1,16] (Figure 1).

A number of scaffolds have been synthesized and evaluated as PDE5 inhibitors such as quinolone [7,17], pyridopyrazinone [1,4,7,18], phthalazine [5], pyrazolopyrmidinone [19,20], β-carboline [6,21], 1,3-benzodioxole [6,22], imidazotriazinone [21,23] and pyrroloquinolone derivatives [24]. Among them is the pyridopyrazinone derivative (compound **4**) [1] (Figure 1). Pyridopyrazinone-based scaffold has a wide spectrum of biological activities in the field of medicinal chemistry. It can act as CRF-R1 antagonist [25,26,27,28,29], PI3K inhibitor [30,31,32] and antiproliferative agent [30,33,34,35,36,37,38]. Pfizer Global R&D described and evaluated a new series of tri-substituted pyridopyrazinone derivatives for their PDE5 inhibitory activities (compounds **t37**–**93**) [1,4] (Appendix A). Recently, we also described the synthesis and biological evaluation of a novel series of mono-substituted pyrido [2,3-*b*]pyrazinone derivatives as anticancer agents (series **A**–**H**) [39,40] (Figure 2, Appendix A). Our goal was to evaluate these candidates (series **A**–**H**) as potential PDE5 inhibitors since they shared the same structural and molecular features of pyridopyrazinone core scaffold in the previously reported PDE5 inhibitors invented by Pfizer Global R&D. In addition, the new structural combination and orientation will be evaluated and compared with that of previously reported PDE5 inhibitors (Figure 3). In the reported PDE5 inhibitors, pyridopyrazinone ring was designed to be the central scaffold in which it was decorated with multiple aryl substitutions (group 1–3, left-hand side in Figure 3) with different size. The new tested compounds were divided into 3 motifs (right-hand side in Figure 3); central linker group (B) and 2 terminal binding motifs (core pyridopyrazinone ring, A; and terminal aryl group, C). The linker group (B) may be flexible spacer with different atom lengths (series **A**–**E**); rigid five-membered rings (series **F** and **G**); and fused cyclic system (series **H**).

## 2. Materials and Methods

### 2.1. Chemistry

The tested compounds were synthesized as reported in our previous publications [39,40]. The experimental method and full characterization for both key intermediates (**IV**, **VII** and **VIII**) and final compounds (series **A**–**H**) are cited in the Appendix A. General methods for the chemical experiments were carried out as previously reported [39,40].

### 2.2. In-Silico Screening

The in-silico screening of the pyridopyrazinone derivatives (training set, **t1**–**150** and test set, series **A**–**H**) was conducted using MOE (Molecular Operating Environment) software [41] through Windows 10 operating system.

#### 2.2.1. 2D-QSAR Analysis

The 2D-QSAR analysis was designed based on the linear regression model. The procedure for generating the 2D-QSAR model was summarized in the following steps:

##### Assembling the Database for the Training Set and Test Set

Both molecular structures of training set (**t1**–**150**) and the test set (series **A**–**H**) were drawn using ChemDraw software. The molecular structures were migrated to MOE software. The migrated molecular structures were 3D protonated and minimized using MOE tools then loaded to separate databases. The experimental PDE5 IC_50_ values of the training set were presented in the training set database.

##### Calculating the 2D-Descriptors

Eighteen 2D-atomic contribution model descriptors were selected and calculated for the training set (log of the octanol/water partition coefficient, SlogP and molar refractivity, SMR)

##### Pruning the Proposed 2D-Descriptors

The calculated set of 2D-descriptors were pruned in order to filter the optimum set for the molecules under consideration (training set). QSAR contingency analysis was conducted to filter the appropriate descriptors for the designed model. A recommended set of 2D-descriptors was defined according to a number of statistical values. Seventeen 2D-descriptors were finally filtered to be used for the proposed 2D-QSAR model.

##### Fitting the Experimental IC_50_ Values

The 2D-QSAR model has been run with partial least squares (PLS) as model fitting procedure and was conducted to the training set in order to fit the dependent variables (IC_50_) to the independent variables (calculated 2D-descriptors) using linear regression analysis.

##### Cross Validating the Designed Model

The 2D-QSAR model was validated using leave-one-out method (LOO) in order to identify the predicted IC_50_ and residuals.

##### Estimating the Predicted IC_50_ of the Test Set

The IC_50_ values of the test set (series **A**–**H**) were predicted using the assembled fit model obtained from the training set analysis. The predicted results were used to build a rational structure activity relationship (SAR).

#### 2.2.2. Molecular Docking

The X-ray structure of phosphodiesterase enzyme type 5 (PDE5) domain with its reference ligand (**t48**) was downloaded from the protein databank RCSB PDB (PDB ID: 3HC8) [4]. The X-ray structure of PDE5 was visualized and manipulated by MOE (Molecular Operating Environment).

The experimental protocol of molecular docking screening is cited in Appendix A and was performed as described in our previous publication [42].

### 2.3. In Vitro Enzyme Assay

Tested compounds (series **B**, **D**, **E** and **G**) were tested against phosphodiesterase enzyme type 5 (PDE5) in order to investigate their inhibitory activities. The in vitro enzyme assay was conducted using Transcreener Fluorescence polarization assay [43] to detect the reaction product GMP. The compounds were suspended in DMSO to make 10 mM stock solutions. The procedure was conducted using the following buffer system; 10 mM Tris, pH 7.5, 5 mM MgCl_2_, 0.01% Brij 35, 1 mM DTT and 1% DMSO. The tested compounds were evaluated in a 10-dose IC_50_ with 3-fold serial dilution starting at 50 uM against methoxyquinazoline as standard control. The tested samples were added into the enzyme by Acoustic Technology. The substrate was added and incubated for 1 h. Fluorescence polarization (FP) was measured after 90 min incubation at rt and mP was calculated. Data were analyzed based on the GMP standard curve.

### 2.4. Molecular Dynamic (MD) Simulation

Protein-ligand complex MD simulation was conducted to validate the molecular docking results. The binding complex of the most potent derivatives among the tested compounds (**11b**) was selected for molecular dynamic analysis. The crystal structure of the human PDE5 in complex with **t48** (PDB ID: 3HC8 [4]) was used as a control.

The experimental protocol of molecular dynamic simulation is cited in the Appendix A and was performed as addressed in our previous publication [44].

## 3. Results and Discussion

### 3.1. In-Silico Screening

Before conducting the in vitro wet-lab analysis for the tested compounds (series **A**–**H**), in-silico screening studies will be performed to predict the target activity as well as to minimize the test set for further in vitro investigation. More than a hundred PDE5 inhibitors were reported in the literature with their IC_50_ values. Therefore, 2D-QSAR analysis was selected as a predictive tool using these reported inhibitors as a training set [45,46]. In addition, the isolated X-ray structure of PDE5 protein, complexed with its inhibitors, has been deposited in a protein data bank. Thus, a molecular docking study can be performed in this study to explore the possible binding mode of the tested compounds inside the active site of PDE5 [47,48]. Consequently, in silico studies, including 2D-QSAR analysis and molecular docking were performed and described in order to determine a number of candidates for further in vitro enzyme assay as PDE5 inhibitors.

#### 3.1.1. 2D-QSAR Analysis

A group of 150 reported derivatives with different scaffolds along with their experimental PDE5 IC_50_ values was used as training set in order to build a rational 2D-QSAR model (**t1**–**150**) (Appendix A). Eighteen 2D-descriptors were built in the proposed model using MOE software [41].

The atomic contribution model descriptors were selected; SlogP descriptors, that calculate the logP (o/w) from the given chemical structures (SlogP_VSA0–9). In addition, SMR descriptors, which assume the correct protonation state of the designed compounds (SMR_VSA0–7) were considered in this study.

QSAR contingency analysis was used to filter the important and the most influential descriptors for the designed QSAR model. Seventeen 2D-descriptors were filtered as the best predicted 2D-descriptors which satisfied the QSAR contingency analysis statistical values (contingency coefficient, Cramer’s V, entropic uncertainty and linear correlation).

The validation of this model showed that 145 compounds had Z-scores values less than 2.5 which revealed that the proposed 2D-descriptors were enough for the designed pharmacophore model. Furthermore, the training set (**t1**–**150**) with their experimental PDE5 IC_50_ were suitable for the model fit. In addition, it showed a suitable values of correlation coefficient of (R^2^ = 0.6084) (Figure 4 and Figure 5) that is agreed to establish the reliability of the QSAR model and to apply this model against the external test set to evaluate the predictive IC_50_ values.

The QSAR model was additionally cross-validated using the Leave-One-Out approach (LOO) and concordance correlation coefficient (CCC) with value of 0.8372.

The predicted PDE5 IC_50_ values of the test set (series **A**–**H**) were estimated using the predicted fit obtained from the training set (Table 1). The results revealed that the compounds involved in the test set showed different predicted inhibitory activity towards phosphodiesterase enzyme type 5 (PDE5) with wide range of IC_50_ values in the nanomolar level.

A number of tested compounds (series **B**, **D**, **E** and **G**) showed promising predicted IC_50_ values in similar range of that of training set (**t1**–**150**). The results showed that the derivatives with flexible linkers with maximum of 4 atoms (series **B**, **D** and **E**) and the rigid five-membered spacer (series **G**) between the pyridopyrazinone scaffold and the terminal phenyl ring may have a promising role in the PDE5 inhibition.

#### 3.1.2. Molecular Docking

The computational approach was carried out to characterize the microscopic interactions and the binding modes between the tested compounds (series **A**–**H**) and the PDE5 active site. Molecular dynamic studies of PDE5 X-ray structure showed that the active site of PDE5 composed of 2 merged pockets; right-hand room (H pocket, hydrophobic pocket) which is close to the M-loop and has a significant number of hydrophobic amino acid residues (Phe 786, Phe 820 and Leu 824) and the left-hand room (Q pocket, cGMP binding site) has the conserved amino acid residue of PDE5 (Gln 817) (Figure 6). In addition, it shed light on the importance of hydrogen bonds with Gln 817, aryl interactions with Phe 786 and Phe 820 for the PDE5 inhibitors [49].

In this study, the three-dimensional complex structure of human PDE5 (PDB ID: 3HC8) [4] with reference ligand (**t48**) was employed to investigate the binding modes of the tested compounds (test set, series **A**–**H**).

In order to obtain reliably binding structures, the molecular simulation of both reference ligand (**t48**, pyridopyrazinone-based ligand) and sildenafil (**1**, FDA approved PDE5 inhibitor) were evaluated in the selected PDE5 protein domain (PDB ID: 3HC8).

The crystal structure of the reference ligand (**t48**) and sildenafil (**1**) agreed well with their docked poses with the lowest docking scores and the root mean square deviation (rmsd) of them. They formed the key hydrogen bond interaction with Gln 817 as well as the important aryl-aryl interactions with Phe 786 and Phe 820 (Figure 7 and Table 2).

Most of the test set showed different interactions with the conserved amino acid residues of the PDE5 active site (Gln 817 and Phe 820).

The results revealed that the tested compounds (flexible-based spacers) shared similar binding modes with those of the reference ligand (**t48**) and sildenafil (**1**).

Moreover, the results showed the crucial nature of the spacer between the 2 binding motifs (pyridopyrazinone and the terminal aryl group) in order to bring these 2 motifs together into the active site of the PDE5 enzyme to occupy both H and Q pockets. For the derivatives with flexible spacers (series **A**–**E**), the generated docking posed showed that these flexible spacers tended to twist to bring the 2 binding motifs into the active site. Where, the pyridopyrazinone ring was directed to the H pocket to afford a number of hydrophobic interactions. While the other terminal aryl group was anchored towards the Q pocket to interact with the key amino acid in this room (Gln 817). However, for the longer spacer (>5 atoms; series **A**), the majority of the generated docking poses showed that there is an exclusive binding of pyridopyrazinone scaffold into the active site leaving the terminal aryl group in the solvent-exposed area. It was suggested that there is an optimum length of the spacer (5 atoms), after which, this spacer may fail to afford a stable twisting to bring the 2 binding motifs together inside the PDE5 binding site.

Series **C** (3 atoms-spacer) was rolled out from our previous suggestion. The bulky size of the terminal aryl groups in this series (**C**) might obstruct the introduction of this bulky motif into its corresponding pocket (Q pocket, Gln 817-containing room).

Series **F** and **G** (with 5 membered ring rigid spacers) showed interesting docking poses into the PDE5 active site. Where, pyridopyrazinone core scaffold as well as the terminal aryl ring were docked into the active site. These rigid spacers were acted like a perfect hinge to keep these 2 binding motifs oriented in such angle to afford the required interactions with the key amino acid residues in PDE5 active site (Figure 8). It was noticed that these rigid spacers in series **F** and **G** (5 membered ring) were retained at the gate of the active site affording hydrophobic interaction with Phe 820 and guiding the 2 binding motifs to the key binding pockets. For series **G**, in particular, the acetyl substituent was anchored towards the solvent-exposed area.

For the fused ring spacers (series **H**), the fused system could not have the relative flexibility found in the other series. The docking results of this series (**H**) showed a limited number of interactions compared to the other series.

### 3.2. In Vitro Enzyme Assay

From in silico docking studies, the tested compounds explored the same key interaction of reported competitive PDE5 inhibitors. Therefore, it was suggested that the tested compounds will inhibit PDE5 protein through competitive binding with the native substrate at the active site. This competitive inhibition would be validated through in vitro enzyme assay. Where, the tested compounds along with the native substrate will be treated with the protein of interest (PDE5) in order to evaluate the inhibition capacity of the tested compounds. Therefore, based on in silico studies results, series **B**, **D**, **E** and **G** were selected to evaluate the in vitro enzyme assay over PDE5 enzyme to identify their IC_50_ values (Table 3).

The results revealed that the tested compounds showed different inhibitory activities over PDE5 enzyme. In addition, both series **E** (hydrazine-based derivatives with 4-atom flexible spacer) and **G** (oxadiazole-based derivatives with 5-membered rigid linker) exhibited potent inhibitory activities compared to other tested derivatives (IC_50_ 18.13–41.41 nM). Moreover, it was shown that compound **11b** (series **G**, oxadiazole-based derivatives with terminal 4-NO_2_ substituted phenyl ring) showed the most promising result among the tested compounds (IC_50_ 18.13 nM). The in vitro enzyme data of series **G** was matched with the molecular docking finding. It was suggested that the potent activity of compound **11b** may be referred to the 5-membered rigid spacer along with the additional HBA group (4-NO_2_) at the terminal aryl group in that compound.

The results revealed that the tested pyridopyrazinone-based scaffold can exhibit enzyme inhibition for PDE5. In addition, HBD/HBA-bearing spacer can play a crucial role in the potency (flexible linker and 5-membered rigid spacer). Moreover, the size and the substitution at the terminal aryl group has a great impact on the inhibitory effect of tested series (phenyl ring decorated with HBD/HBA groups) (Figure 9 and Figure 10).

### 3.3. Molecular Dynamic (MD) Simulation

The aforementioned docking simulation study predicted and confirmed the potent inhibitory activity of compound **11b** with PDE5 with several favorable interactions. However, molecular docking only considers flexible ligand conformations whereas the protein is usually kept in a rigid state [50]. Hence, to evaluate the binding pose stability and protein conformation dynamics, a molecular dynamics (MD) simulation study of the protein (PDE5)-ligand (**11b**) complex was carried out for 50 ns. Both PPDE5 protein and ligand root mean square deviation (RMSD) calculation, protein root mean square fluctuation (RMSF), radius of gyration (Rg) analysis, and H-bond occupancy calculations were conducted for the trajectory analysis. The molecular dynamic simulation was validated by monitoring the RMSD calculations. The rigid and flexible sections of the protein structure were identified using the RMSF method. It is well-known that the standard measure of a molecule’s deviation from its initial position is defined as the RMSF. Finally, the radius of gyration was used to estimate the folding properties and compactness of the protein–ligand complexes (Rg) [44]. The results of the MD study revealed that all of the protein residues possess an RMSF value below 2 Å (Figure 11A) which validates that the ligand (**11b**) stabilized the amino acid positions. Throughout the simulation, the ligand-protein complex exhibited average RMSD value of 2.8 Å (Figure 11B) and the PDE5 protein did not show drastic unfolding with Rg around 1.97 ± 0.2 nm (Figure 11C) indicating the stability of the formed complex and the validity of compound **11b** as a promising PDE5 inhibitor. The ligand **11b** also established a minimum of two hydrogen bonds (HBs) throughout the simulation, and occasionally exhibited 4 HBs with PDE5 protein structure (Figure 11D).

## 4. Conclusions

In conclusion, several pyridopyrazinone derivatives were investigated as phosphodiesterase enzyme type 5 (PDE5) based on in-silico screening and in vitro enzyme assay. 2D-QSAR analysis showed that the test set had a predicted inhibitory activity towards PDE5 in nanomolar level. Molecular docking showed that most of tested compounds exhibited a number of interactions with conserved amino acids of PDE5 active site (Phe 786, Phe 820 and Gln 817) similar to that of sildenafil. Compound **11b** (series **G**, oxadiazole-based derivatives with terminal 4-NO_2_ substituted phenyl ring and rigid linker) showed the most promising results among the tested compounds in both in silico studies (S = −7.0486 kcal/mol) and in vitro PDE5 enzyme assay (IC_50_ 18.13 nM). Finally, MD simulations were performed to validate the stability of compound **11b** into PDE5 binding site. The scaffold reprofiling approach afforded pyrido [2,3-*b*]pyrazinone scaffold, in combination with the crucial rigid linker, as a potential therapeutic potential to act as PDE5 inhibitor. In addition, the study revealed the promising potency and the stability of compound 11b with structural simplification compared to the other reported pyridopyrazinone derivatives with trisubstituted motifs (compound **4**), which may open the door for the applicability of this particular compound (**11b**) as a promising preclinical candidate.

## Figures and Tables

**Figure 1 pharmaceutics-14-01954-f001:**
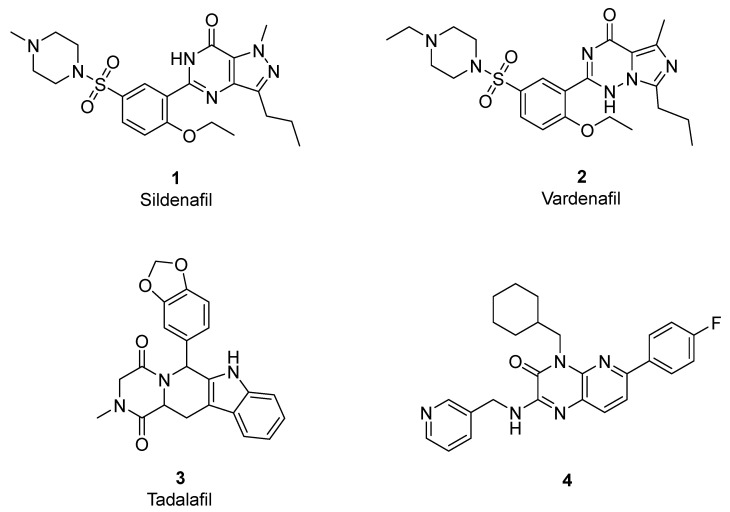
Reported PDE5 inhibitors.

**Figure 2 pharmaceutics-14-01954-f002:**
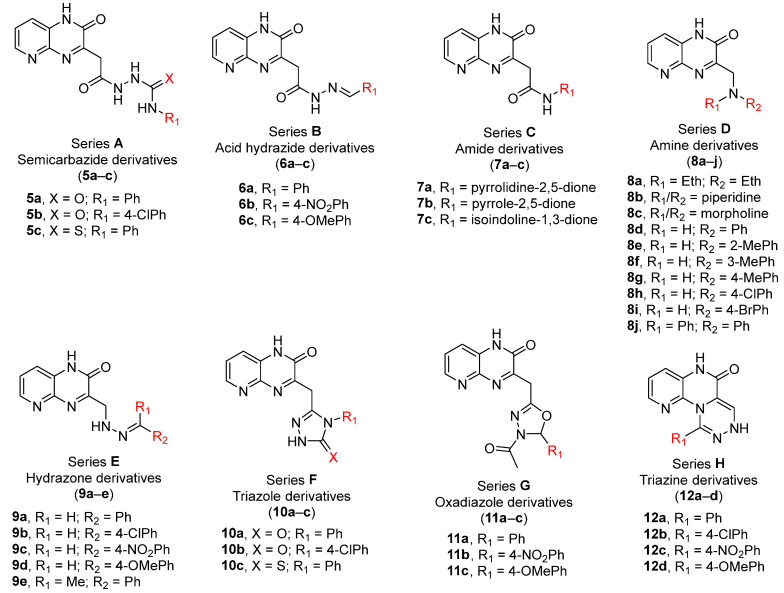
The tested pyridopyrazinone derivatives. Eight different series (**A**–**H**) with structural variation in the spacers between the core pyridopyrazinone scaffold and the terminal aryl motif. Series **A**–**E** with flexible spacer with different length as well as different incorporated HBD/HBA atoms. Series **F** and **G** with 5-membered rigid spacers (triazole and oxadiazole, respectively). Series **H** with rigid tricyclic system. (Appendix A).

**Figure 3 pharmaceutics-14-01954-f003:**
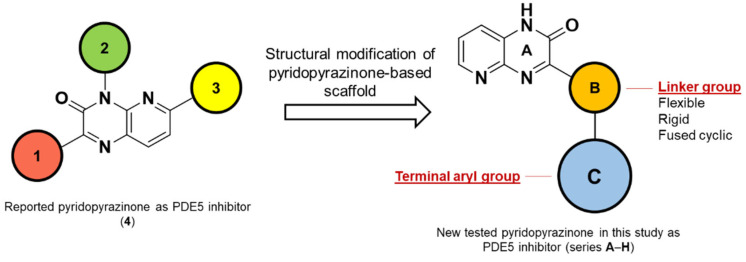
Rational structural modification of the reported pyridopyrazinone-based derivatives as PDE5 inhibitors. The orientation of the terminal groups is modified structurally. In the reported inhibitors, pyridopyrazinone scaffold is designed to be the central motif. In the tested compounds, the pyridopyrazinone ring is functioning as one of binding motifs.

**Figure 4 pharmaceutics-14-01954-f004:**
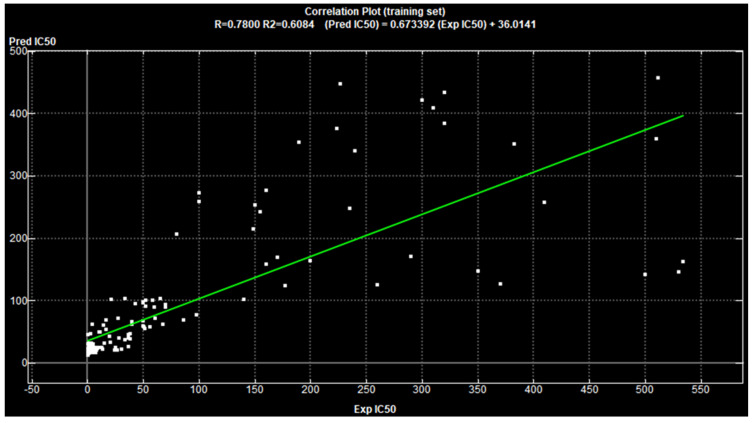
Correlation plot of the training set (**t1**–**t150**) showing R^2^ value = 0.6084.

**Figure 5 pharmaceutics-14-01954-f005:**
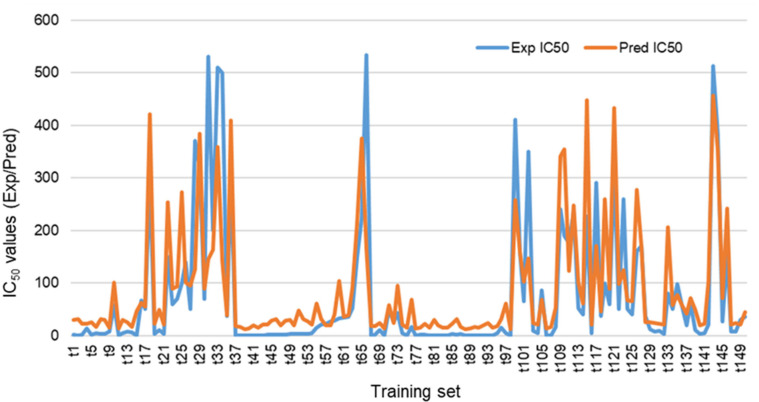
Overview of the graphical presentation of PDE5 IC_50_ of training set (**t1**–**150**) shows correlation between the experimental (blue color) and predicted (red color) IC_50_ values.

**Figure 6 pharmaceutics-14-01954-f006:**
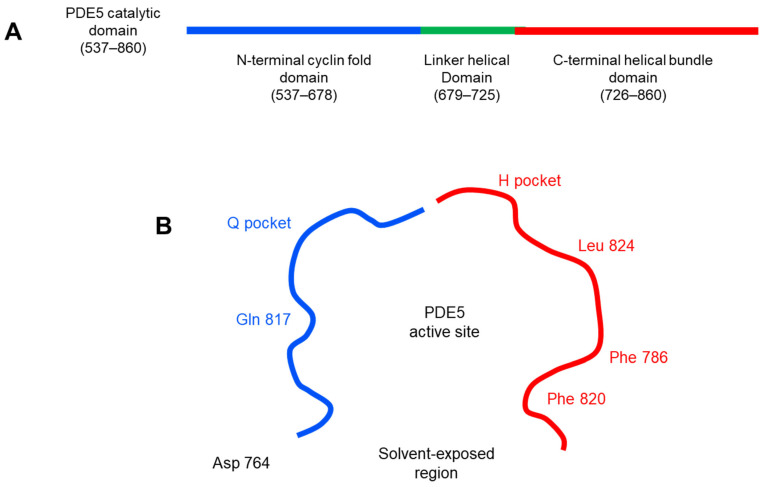
2D diagram of PDE5 active site (PDB ID: 3HC8). (**A**) the catalytic domain of PDE5 enzyme can be divided into three subdomains: N-terminal cyclin-fold domain (residues 537–678 in blue), linker helical domain (residues 679–725 in green) and C-terminal helical bundle domain (residues 726–860 in red). (**B**) the active site of PDE5 can be described in 3 core regions: H pocket (hydrophobic pocket) in red color with key hydrophobic amino acid residues (Phe 786, Phe 820 and Leu 824), Q pocket (cGMP binding site) in blue color with Gln 817 as the key amino acid residue in that site and solvent-exposed region with Asp 764.

**Figure 7 pharmaceutics-14-01954-f007:**
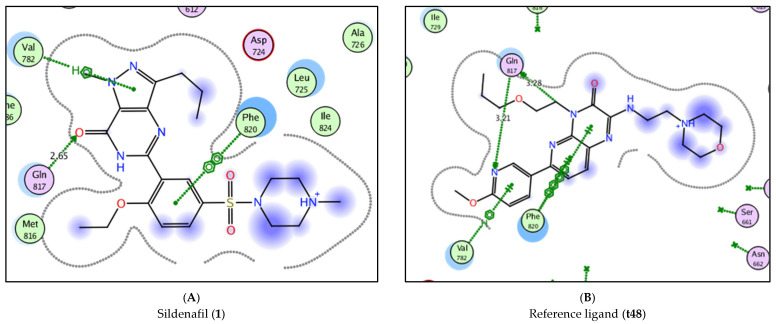
2D interaction of sildenafil (**1**, **A**) and reference ligand (**t48**, **B**) with PDE5 protein domain (PDB ID: 3HC8).

**Figure 8 pharmaceutics-14-01954-f008:**
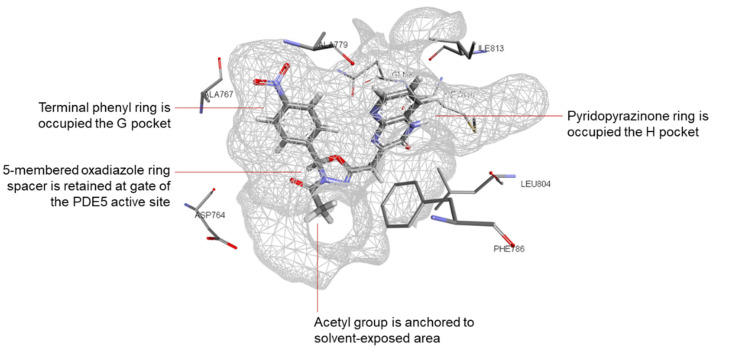
3D interaction of the docked pose of compound **11b** (series **G**, 5-membered-based linker) with PDE5 binding site (PDB ID: 3HC8). The oxadiazole linker is retained at the active site gate and acts as a hinge between the 2 binding motifs of compound **11b**. The pyridopyrazinone ring is oriented into the H pocket and affords hydrophobic interactions (Phe 786 and Phe 820) and H bonding with Met 816 (3.45 Å). The other terminal binding motif (4-NO_2_-phenyl ring) is directed to the G pocket and affords a significant H bonding with Gln 817 (2.69 Å). The acetyl substitution is waved in the solvent-exposed area.

**Figure 9 pharmaceutics-14-01954-f009:**
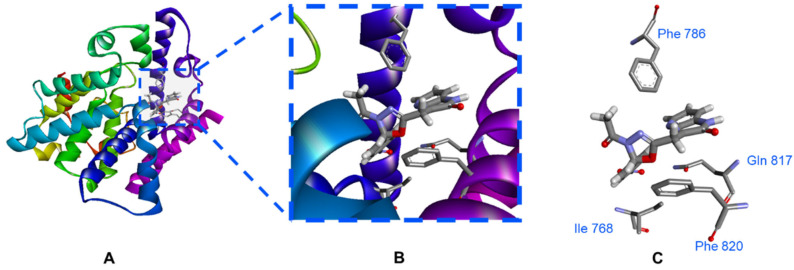
The expected binding interaction between the most active compound (**11b**) and PDE5 protein (PDB ID: 3HC8). (**A**) the 3D conformation of PDE5 protein with the allocated compound **11b** in the active site; (**B**) the condensed view of the PDE5 active site showed the key amino acid residues and their close distance to compound **11b** binding motifs (pyridopyrazinone scaffold and terminal aryl group); (**C**) the isolated view of PDE5 active site showed compound **11b** with significant interactions with the key amino acid residues in the PDE5 active site (Phe 768, hydrophobic interaction with terminal phenyl ring; Phe 786, hydrophobic interaction with pyridopyrazinone ring; Gln 817, HB interaction with 4-NO_2_ group; and Phe 820, hydrophobic interactions with both central oxadiazole ring and pyridopyrazinone ring).

**Figure 10 pharmaceutics-14-01954-f010:**
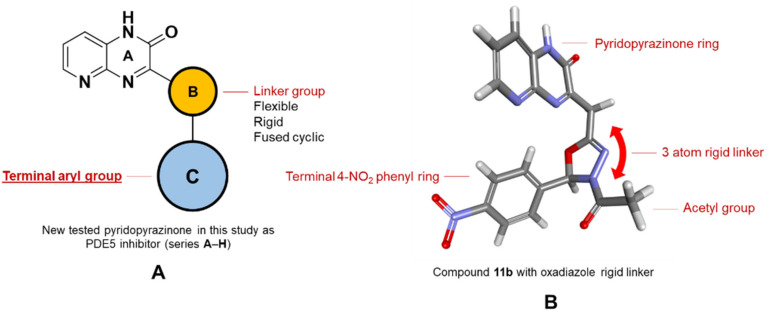
The proposed structure activity relationship (SAR) of the new pyridopyrazinone-based derivatives (series **A**–**H**) into PDE5 protein (PDB ID: 3HC8). (**A**) pyridopyrazinone ring favors to be anchored into the H pocket close to M-loop and affords hydrophobic interactions as well as HB interaction. The central linker group acts as a hinge between the 2 binding motifs. The flexible spacer with optimum length (5 atoms) will allow this hinge to bring the 2 binding motifs inside the binding pocket. The rigid spacer with five-membered ring is rigid enough to keep the angular orientation between the 2 binding motifs inside the active site. The fused cyclic spacer creates an unfavorable strain between the 2 binding motifs. The terminal aryl motif with HBD/HBA groups will enhance the binding through HB to Gln 817 in the Q pocket. Bulky aryl groups (series **C**) will deteriorate the residence of the 2 binding motifs together into the active site. (**B**) the stable conformation of compound **11b** inside PDE5 binding site. The oxadiazole rigid ring affords a stable hinge (3 atoms length) between the 2 binding motifs. The terminal NO_2_ group affords a strong HB interaction with Gln 817 in Q pocket (2.69 Å). The acetyl substitution swims in the solvent-exposed area.

**Figure 11 pharmaceutics-14-01954-f011:**
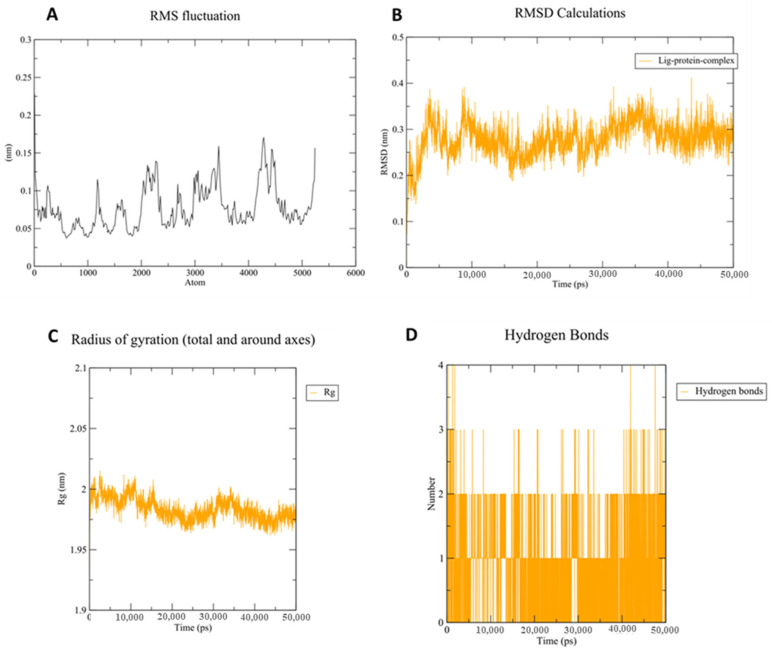
Molecular dynamic (MD) analysis of ligand **11b** into PDE5 protein (PDB ID: 3HC8). (**A**) RMSF values of PDE5 residues during 50 ns MD simulations; (**B**) RMSD of ligand (**11b**) and protein (PDE5) backbone obtained from 50 ns MD simulation; (**C**) radius of gyration (Rg) showing no drastic folding of PDE5 during the 50 ns MD simulation; (**D**) assessment of the number of H-bonds formed between the ligand (**11b**) and the target (PDE5) during 50 ns MD simulations.

**Table 1 pharmaceutics-14-01954-t001:** The predicted PDE5 IC_50_ values of the test set (series **A**–**H**) in nM.

Compound	Predicted IC_50 (nM)_	Compound	Predicted IC_50 (nM)_
**5a**	202.71	**8i**	71.38
**5b**	262.99	**8j**	137.24
**5c**	147.43	**9a**	39.33
**6a**	51.38	**9b**	99.61
**6b**	70.36	**9c**	58.31
**6c**	7.83	**9d**	−4.21
**7a**	90.20	**9e**	66.24
**7b**	120.98	**10a**	125.77
**7c**	147.98	**10b**	186.05
**8a**	27.49	**10c**	70.48
**8b**	26.75	**11a**	49.54
**8c**	38.97	**11b**	68.52
**8d**	81.56	**11c**	6.00
**8e**	66.27	**12a**	111.41
**8f**	66.27	**12b**	171.69
**8g**	66.27	**12c**	130.39
**8h**	141.84	**12d**	67.86

**Table 2 pharmaceutics-14-01954-t002:** Binding interactions and docking scores of sildenafil (**1**), reference ligand (**t48**) and the test Scheme 5 active site.

Compound	Docking Score (S) ^a^	Hydrophobic Interactions	Hydrogen Bond (Å)
**1**	−8.3805	Val 782, Phe 820	Gln 817 (2.65)
**t48**	−9.8829	Val 782, Phe 820	Gln 817 (3.21), Gln 817 (3.28)
**5a**	−6.7026	-	-
**5b**	−7.4526	Leu 804	-
**5c**	−6.6958	His 613	Asp 764 (2.83)
**6a**	−6.3718	His 613	-
**6b**	−6.9287	Phe 820	His 685 (3.40), Asp 764 (3.19), Gln 817 (2.88)
**6c**	−6.6302	-	-
**7a**	−6.4060	Leu 804, Phe 820	Leu 765 (3.55)
**7b**	−6.3842	Val 782, Phe 820	-
**7c**	−6.4797	His 613	-
**8a**	−5.9536	-	Gln 817 (3.65)
**8b**	−6.4089	Val 782, Phe 820	Gln 817 (2.99)
**8c**	−6.0430	Val 782	Gln 817 (2.97)
**8d**	−6.0887	-	-
**8e**	−6.3671	Phe 820	Gln 817 (2.88)
**8f**	−6.5098	Val 782	Gln 817 (2.89)
**8g**	−6.4050	Leu 804, Phe 820	-
**8h**	−6.2434	His 613, Phe 820	Gln 817 (3.10)
**8i**	−6.5529	Leu 804, Phe 820	-
**8j**	−7.1926	-	-
**9a**	−6.4902	Val 782, Phe 820	Gln 817 (3.48)
**9b**	−6.6590	-	-
**9c**	−6.4239	-	Tyr 612 (3.20)
**9d**	−6.9129	Leu 804	-
**9e**	−6.8623	Val 782, Phe 820	Gln 817 (3.33)
**10a**	−6.7878	-	-
**10b**	−6.6090	-	-
**10c**	−6.7319	-	Tyr 612 (3.33), Asp 764 (2.96)
**11a**	−6.8508	-	-
**11b**	−7.0486	Phe 786, Phe 820	Met 816 (3.35), Gln 817 (2.69)
**11c**	−7.4782	Ile 768, Leu 804	-
**12a**	−6.1199	Val 782	Gln 817 (3.35)
**12b**	−6.1236	Phe 820	-
**12c**	−6.4532	Phe 820	-
**12d**	−6.6629	Phe 786, Phe 820	-

^a^ kcal/mol.

**Table 3 pharmaceutics-14-01954-t003:** The experimental and predicted PDE5 IC_50_ values of the test set (series **B**, **D**, **E** and **G**) in nM.

9	IC_50 (nM)_		Compound	IC_50 (nM)_	
	Exp	Pred		Exp	Pred
**6a**	59.13	51.38	**8i**	137.31	71.38
**6b**	67.91	70.36	**8j**	ND	137.24
**6c**	47.00	7.83	**9a**	32.20	39.33
**8a**	93.83	27.49	**9b**	30.42	99.61
**8b**	44.63	26.75	**9c**	41.41	58.31
**8c**	89.01	38.97	**9d**	32.34	−4.21
**8d**	101.32	81.56	**9e**	29.40	66.24
**8e**	ND	66.27	**11a**	26.33	49.54
**8f**	21.01	66.27	**11b**	18.13	68.52
**8g**	ND	66.27	**11c**	31.03	6.00
**8h**	ND	141.84			

Exp, experimental; Pred, predicted; ND, not determined.

## Data Availability

Data is contained within the article and Appendix A.

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
