# Peer review of "Scaffold Repurposing Reveals New Nanomolar Phosphodiesterase Type 5 (PDE5) Inhibitors Based on Pyridopyrazinone Scaffold: Investigation of In Vitro and In Silico Properties"

_pharmaceutics, 2022, doi:10.3390/pharmaceutics14091954_

Round 1

Reviewer 1 Report

The manuscript entitled “Drug Repurposing Reveals New Nanomolar Phosphodiesterase 2 Type 5 (PDE5) Inhibitors Based on Pyridopyrazinone Scaffold: 3 Investigation of In Vitro and In Silico Properties) by Kamelia et. al.  describes the effect of already made anticancer agents on PDE5 applying the Drug Repurposing strategy. In general the manuscript is well written and the strategy was successful as the results revealed. The number of compounds was good to build SAR and the in vitro results support the modeling work. The manuscript can be published with some minor comments.

Decision: minor revision

Comments :

1- The author should start numbering all the structures from 1 in figure 1 so sildenafil will take number 1 and so on

2- In all modeling figures please state the PDB code used below the figure not only in the text

3- In the in vitro result table, it will be good to add a new column contains the predicted IC50

Author Response

We appreciate the time and effort that you have dedicated to provide your valuable feedback on this manuscript. Please check the attached file.

Reviewer 2 Report

The manuscript - pharmaceutics-1872916-peer-review-v1.pdf- “Drug Repurposing Reveals New Nanomolar Phosphodiesterase Type 5 (PDE5) Inhibitors Based on Pyridopyrazinone Scaffold: Investigation of In Vitro and In Silico Properties”

This study may be relevant and I consider it will be of interest to the scientific community. In my opinion, if the mentioned major issues are solved the manuscript could meet the journal's requirements.

In the following, I have included only a few points that support the major shortcomings of this manuscript.

Reviewer questions:

1. In the title, the authors specify “Drug Repurposing…” For me it is not very clear which drugs were repurposed because the conclusion is “Compound 7b … showed the most promising results among the tested compounds in …”

2. I do not agree with the author’s remark (lines 126-127) that “In addition, it showed a suitable values of correlation coefficient of (r2 = 0.2444)”. I have serious doubts about the statistical significance and predictive capacity of the model. In general, a QSAR model is acceptable when it has an r2 value greater than 0.6 (see, Golbraikh and Tropsha’s works). In addition, for a robust and predictive QSAR model various statistical parameters must be calculated: E.g.  (i) Internal validation ((e.g. concordance correlation coefficient (CCC) for the training set (values greater than 0.85 must be obtained), squared correlation coefficient of randomized models, R2scr, squared leave-one-out correlation coefficient of randomized models, Q2scr (if the Y-axis intercept of the regression line does not exceed 0.3–0.4 for R2scr, and 0.05 for Q2scr, the model is considered free of chance correlation). (ii) External validation (e.g. concordance correlation coefficient (CCC) for prediction set (values greater than 0.85 must be obtained), Q2F1, Q2F2, Q2F3 (values greater than 0.7 must be obtained). (iii) The applicability domain (AD) is important too, to check prediction reliability and to produce reliable predicted data for chemicals not included in the training set. The results of the external validation process show the applicability domain (AD) of the QSAR model and, therefore, the robustness of the model to predict the activity of new molecules. The structural applicability domain of the QSAR final equations should be checked using the Insubria graph; etc.

3. In the Methodology section, line 336 and 352,  it is specified “…100 ns…”, but in the Results section, line 229 and 247, it is specified “…was carried out for 50 ns.”

4. The main goal of each research is to find new candidates with improved profiles compared to those currently on the market. I don’t find such kinds of information, here. Please investigate the obtained results by comparing them with other approved drugs.

5. I encourage the authors to address all the recommendations, to improve the quality of the manuscript that will surely be able to meet the requirements of the journal publication, and to resubmit it after the revision. 

Author Response

(The authors gave the same response as above.)

Round 2

Reviewer 2 Report

The authors addressed the main concerns from the reviews, the revised version of the manuscript appears to be good.

I have one important remark:

The authors' answer to point 1: “Response: (N.B, compound 7b has been renumbered to 11b). An additional sentence was mentioned in the conclusion section to emphasize the drug reprofiling approach of previously reported pyridopyrazinone to act as PDE5 inhibitors using in silico and in vitro screening. Compound 11b was the most active compound and the authors recommended this compound to be the potential candidate in this study.”

So, the question remains. What drugs were repurposed in the present work? Drug repurposing involves exploring new medical uses for preapproved drugs or existing drugs….
